# Evaluating Maize Genotype Performance under Low Nitrogen Conditions Using RGB UAV Phenotyping Techniques

**DOI:** 10.3390/s19081815

**Published:** 2019-04-16

**Authors:** Ma. Luisa Buchaillot, Adrian Gracia-Romero, Omar Vergara-Diaz, Mainassara A. Zaman-Allah, Amsal Tarekegne, Jill E. Cairns, Boddupalli M. Prasanna, Jose Luis Araus, Shawn C. Kefauver

**Affiliations:** 1Integrative Crop Ecophysiology Group, Plant Physiology Section, Faculty of Biology, University of Barcelona, 08028 Barcelona, Spain; luisa.buchaillot@gmail.com (M.L.B.); adriangraciaromero@hotmail.com (A.G.-R.); omarvergaradiaz@gmail.com (O.V.-D.); jaraus@ub.edu (J.L.A.); 2AGROTECNIO (Center for Research in Agrotechnology), Av. Rovira Roure 191, 25198 Lleida, Spain; 3International Maize and Wheat Improvement Center, CIMMYT Southern Africa Regional Office, P.O. Box MP163 Harare, Zimbabwe; Z.mainassaraAbdou@cgiar.org (M.A.Z.-A.); a.tarekegne@cgiar.org (A.T.); J.Cairns@cgiar.org (J.E.C.); 4International Maize and Wheat Improvement Center (CIMMYT), P.O. Box 1041 Nairobi, Kenya; b.m.prasanna@cgiar.org

**Keywords:** maize, nitrogen, phenotyping, remote sensing, Africa, RGB, UAV, CIELab

## Abstract

Maize is the most cultivated cereal in Africa in terms of land area and production, but low soil nitrogen availability often constrains yields. Developing new maize varieties with high and reliable yields using traditional crop breeding techniques in field conditions can be slow and costly. Remote sensing has become an important tool in the modernization of field-based high-throughput plant phenotyping (HTPP), providing faster gains towards the improvement of yield potential and adaptation to abiotic and biotic limiting conditions. We evaluated the performance of a set of remote sensing indices derived from red–green–blue (RGB) images along with field-based multispectral normalized difference vegetation index (NDVI) and leaf chlorophyll content (SPAD values) as phenotypic traits for assessing maize performance under managed low-nitrogen conditions. HTPP measurements were conducted from the ground and from an unmanned aerial vehicle (UAV). For the ground-level RGB indices, the strongest correlations to yield were observed with hue, greener green area (GGA), and a newly developed RGB HTPP index, NDLab (normalized difference Commission Internationale de I´Edairage (CIE)Lab index), while GGA and crop senescence index (CSI) correlated better with grain yield from the UAV. Regarding ground sensors, SPAD exhibited the closest correlation with grain yield, notably increasing in its correlation when measured in the vegetative stage. Additionally, we evaluated how different HTPP indices contributed to the explanation of yield in combination with agronomic data, such as anthesis silking interval (ASI), anthesis date (AD), and plant height (PH). Multivariate regression models, including RGB indices (R^2^ > 0.60), outperformed other models using only agronomic parameters or field sensors (R^2^ > 0.50), reinforcing RGB HTPP’s potential to improve yield assessments. Finally, we compared the low-N results to the same panel of 64 maize genotypes grown under optimal conditions, noting that only 11% of the total genotypes appeared in the highest yield producing quartile for both trials. Furthermore, we calculated the grain yield loss index (GYLI) for each genotype, which showed a large range of variability, suggesting that low-N performance is not necessarily exclusive of high productivity in optimal conditions.

## 1. Introduction

Maize is the most commonly cultivated cereal in Africa in terms of land area and production [1]. Low yields in this region are largely associated with drought stress, low soil fertility, weeds, pests, diseases, low input availability, low input use, and inappropriate seeds [2]. After water, nitrogen (N) is the single most important input for maize production, and the lack of N is considered to be the principal constraint to cereal yields in areas with more than 400 mm of average annual rainfall in Sub-Saharan Africa (SSA) [3], but fertilizer application in SSA is negligible, accounting for less than 1% of the global N fertilizer application [4]. As such, efforts to increase maize production capacities in low fertilizer conditions may contribute substantially to improving food security and well-being in the region [5]. One of the strategies considered for increasing maize yield with regards to N inputs in SSA is breeding to improve yield under nutrient deficiency or towards specific adaptation to increase performance under low-nitrogen conditions. Furthermore, the adaptation of maize to lower fertilizer conditions may improve agricultural economics at equal or even better levels of production with lower required inputs, less runoff, and resource extraction that may additionally result in reducing environmental degradation and the loss of ecosystem services [6,7,8].

Plant scientists, especially breeders and agronomists, face the challenge of solving these limitations while considering the additional implications of climate change on food security [2,9]. In that sense, affordable technologies capable of monitoring crop performance, improving yield prediction, or assessing phenotypic variability for breeding purposes are aimed at surpassing the bottlenecks in the way of full exploitation of this technology [10,11]. One of the first non-destructive and analytical tools was the chlorophyll meter, based on radiation absorbance by leaves in the red and near-infrared regions (usually at 650 and 940 nm). These leaf level relative chlorophyll content readings have an indirect and close relationship with leaf N and total chlorophyll concentrations [12,13]. Portable meters have been used for some time on crops as a fairly quick and reliable method for N management [14,15], but this technique is relatively slow compared with newer imaging techniques and does not include the whole plot, thereby capturing less variability than full canopy remote sensing techniques [16].

Remote sensing has become an important tool in the modernization of field-based high throughput plant phenotyping (HTPP), including improvements in yield potential, adaptation to abiotic stressors (drought, extreme temperatures, salinity), biotic limiting conditions (susceptibility to pests and diseases), and even quality traits [5,10,17]. Traditionally, the primary platforms used to obtain remote images of the Earth’s surface were satellites and piloted aircrafts, but these instruments generally do not deliver data at adequate spatial and temporal resolutions necessary for more detailed agricultural applications, such as plant phenotyping [18]. Currently, these limitations can be overcome using more flexible unmanned platforms, such as unmanned aerial vehicles (UAVs), also called remotely piloted aircraft systems (RPASs) or unmanned aircraft systems (UASs) [19,20]. UAVs allow for many quick, precise, and quantitative observations at improved spatial and temporal resolutions and at lower costs with respect to airborne platforms or satellites.

The classical approach of remote sensing platforms, including UAVs, has involved the use of multispectral sensors and the calculation of different vegetation indices associated with plant physiological parameters, such as plant pigments, vigor, and above-ground biomass. In this sense, visible and near-infrared (VNIR) imaging spectroscopy has demonstrated a fairly reliable capacity in biophysical crop assessments in agriculture [21,22,23,24,25,26,27]. For example, the normalized difference vegetation index (NDVI) [28] is a well-known, broadband vegetation index derived from visible and near-infrared reflectance that is closely related to vegetation presence or vigor [29,30]. It can also be measured at the ground level with active portable sensors (e.g., GreenSeeker). Other examples use narrow-band reflectance values for more precise measurements [31] and are often found to be correlated with grain yield and provided reliable information for yield forecasting [32] or specific biophysical properties, such as canopy water content [33] or photosynthetically active pigments, [23,34,35] but require more advanced sensor technologies for adequate quantification. Similarly, thermal infrared (TIR) imaging enables rapid remote observations of plant water status via their cooling capacity and stomatal conductance [36,37]. 

As a low-cost alternative, and at an order of magnitude less expensive than scientific multispectral VNIR or TIR sensors, various red–green–blue vegetation indices (RGB VIs), calculated from commercial RGB cameras, have demonstrated their ability to predict grain yield, quantify nutrient deficiencies, and measure disease impacts [38,39]. With respect to these commercially available RGB cameras, color calibration quality should be assessed prior to scientific use for checking and/or correcting variations in RGB color values, as illumination conditions may influence the accuracy of color reproduction [40,41]. On the other hand, when these types of camera are used for producing multi-image mosaics, within image vignetting should also be assessed, as brightness attenuates away from the image center and appears as artifacts in the image mosaics [42]. Still, RGB VIs can accurately quantify different properties of color and have often demonstrated performance levels similar to or better than NDVI [39]. RGB images can be processed using comparisons between red, green, and blue light broadband reflectance values or through the use of alternate color spaces, as with the Breedpix code suite [43]. The treatment of R, G, and B as separate spectral bands allows for the calculation of the triangular greenness index (TGI), which estimates chlorophyll concentration in leaves and canopies [1], and the normalized green–red difference index (NGRDI), which compares the differences between the green and red bands in a calculation similar to NDVI but with less marked differences and less signal saturation. In the hue–saturation–intensity (HSI) color space, where the hue (H) component describes color chroma traversing the visible spectrum in the form of an angle between 0° and 360°. Thus, the index green area (GA) is the percentage of pixels in the image in the hue range from 60° to 180°, ranging from yellow to bluish green, while the greener green area (GGA) includes a more restrictive range of hue from 80° to 180°, excluding yellowish-green tones that might be partially stressed or senescent. Hence comes the crop senescence index (CSI), which combines GA and GAA to provide a strong discrimination between tolerant and susceptible genotypes in various treatments [44,45]. 

To the end of better quantifying leaf pigment loss, and therefore color changes, due to nitrogen deficiency [46,47], further investigation of the capacities and techniques for accurate color quantification using digital images indicates that there are newer more advanced color models currently in use by photography professionals. In the Commission Internationale de I´Edairage (CIE), CIELab color space model, dimension L* represents lightness; the a* component expresses green to red, with a more positive value representing red, and a more negative value indicating green; and the b* component expresses blue to yellow, in which positive values are towards yellow, and negative values are closer to blue. Correspondingly, in the CIELuv color space model, dimensions u* and v* are perceptually uniform coordinates, where L is again lightness and u* and v* represent axes similar to a* and b* in separating the color spectrum, respectively. For more specific details on the development of these alternate color space RGB indices and their respective transformations, please see [48]. Both CIELab and CIELuv include color calibration corrections through the separation of the color hue from the illumination components of the input RGB signal; for that reason, we have developed two new vegetation indices using these color spaces in a way similar to the conceptual basis for NDVI, using the normalized difference between a* and b* (NDLab) and the normalized difference between u* and v* (NDLuv). Thus, the CIElab and CIEluv color spaces offer the ability to simultaneously contrast green vegetation quantity with both the reddish/brown soil background (fractional vegetation cover or plant growth) and yellowing caused by chlorosis (loss of foliar chlorophyll)—both common symptoms of nitrogen deficiency. Previously, RGB VIs have been employed at both the canopy and at the leaf levels for precise crop management or as effective HTPP techniques in breeding programs aimed to improve crop performance under a wide range of conditions [10]. 

In the research presented here, the RGB VIs described above, namely hue, a*, b*, GA, GGA, NGRDI, and the new NDLab and NDLuv, are examined for their potential as affordable HTPP tools to accurately phenotype commercial and pre-commercial maize genotypes under low- and optimal-N conditions. Firstly, we provide some maize genotype performance comparisons between the low-N and optimal growing conditions in order to provide some initial insights on the potential of selecting for low-N-adapted maize genotypes. Then, we evaluate the performance of a set of remote sensing RGB VIs from natural color images acquired at the ground level and from a UAV platform compared with the performance of the field-based NDVI and SPAD sensors. Additionally, we evaluated how these different sets of plant phenotyping data contribute to improving multivariate model estimations of crop yield in combination with traditional agronomic field data, such as anthesis silking interval (ASI), anthesis data (AD), plant height (PH), and canopy senescence (SEN) in order to determine the level of improvements over traditional practices that they may provide.

## 2. Materials and Methods

### 2.1. Plant Material and Growing Conditions

Field trials for managed low-nitrogen and optimal fertilizer conditions were conducted at the International Center for Maize and Wheat Improvement (CIMMYT) regional station located in Harare, Zimbabwe (−17,800 S, 31,050 E, 1498 m.a.s.l.) (Figure 1). The soil of the station is characterized by a pH slightly below 6, with low managed nitrogen (LOW) treatment for all plots at 25–35% less N compared with the optimal standard fertilization application of 200 kg/ha, here defined as the optimum nitrogen (OP) according to established standard CIMMYT protocols [17]. A set of 49 new maize genotypes that were developed at CIMMYT and 15 commercial maize genotypes in Zimbabwe were selected for the study (Table A1). Seeds were sown during the wet season, on 16 December 2015, in two rows per plot; the rows were 4 m long and 75 cm apart (5.25 m^2^/plot), with 14 planting points per row and 25 cm between the plants within a row. The experiment was carried out in 192 plots with 3 replicates per variety. Both trials were rainfed only, being grown in the Zimbabwe rainy season, with local weather station data recording growing season mean temperature, humidity, and total rainfall of 26°C, 68%, and 700 mm, respectively, effectively eliminating any chances of water stress even without irrigation. 

The trials were harvested in mid-May of 2016, discarding 2 plants at each row end and harvesting the central 3.5 m of each row in order to reduce edge effects. Thus, the total harvested weight corresponded to an area of 5.25 m^2^ (0.75 m apart × 2 rows × 3.5 m long), consisting of the same number of plants per plot (excepting locations of mortality). The cobs processed, and grains dried to approximately 12.5% moisture, such that grain yield (GY, t·ha^−1^) was calculated as follows, where X is the grain weight per plot: GY = (X (kg/plot) *10)/(5.25 m^2^)(1)

The grain yield loss index (GYLI) as the stress index was calculated as:GYLI = (GY at OP − GY at LOW)/(GY at OP) × 100(2)where GY at OP represents the potential grain yield in optimum-nitrogen conditions and GY at LOW corresponds to grain yield in low managed nitrogen conditions [38].

### 2.2. Agronomic Parameters

PH was measured on 19 February 2016 as the length from the soil surface to the base of the tassel (excluding tassel length) using a ruler [49,50] on two representative plants per plot before all plants were hand harvested and grain yield was assessed. ASI was determined by the number of days from sowing until 50% of plants extruded anther AD and the number of days from sowing until 50% of the plants show silks (silking date, SD), such that SD − AD = ASI. SEN was measured visually on a plot basis as the proportion of green leaves 2–5 weeks after anthesis on a 0–100 scale, where 0 = 0% canopy senescence and 100 = 100% canopy senescence (Equation (3)). That technique is based on the different color classes, and given that any part of a leaf with yellow or brown (dry) color was classified as undergoing or having succumbed to senescence, a senescence index was proposed as the ratio between senesced canopy and the total canopy cover:SEN = (YC + DC)/(YC + DC + GC)(3)where GC is green canopy cover, YC is yellow canopy cover, and DC is dry canopy cover [17,44,51,52]. This was measured 4 times during the experimental trial, but only the last SEN measurements from 5 April 2016 were used in this study.

### 2.3. Proximal and Aerial Data Collection

RGB remote sensing evaluations were performed on young maize plants (less than 5 leaves) on 28 January 2016, during the last week of January. For ground RGB VIs, vegetation indices were derived from one picture taken at the ground level for each plot (covering 40–50% of each plot), and UAV RGB VIs were derived from whole plot coverage from the UAV RGB aerial image mosaic of the whole study area as shown in Figure 2. At the ground level, one digital photograph was taken per plot with an Olympus OM-D E-M10 Mark III (Olympus, Tokyo, Japan), holding the camera at about 80 cm above the plant canopy in a zenithal angle and focused near the center of each plot. The images were acquired with a resolution of 16 megapixels with a Micro Four Thirds (M4/3) Live MOS sensor with a focal length of 14 mm, at a speed of 1/125 s with the aperture programmed in automatic mode at a resolution of 4608 × 3072 for a Ground Sample Distance (GSD) of 0.03 cm/pixel. RGB aerial images were acquired using an UAV (Mikrokopter OktoXL, Moormerland, Germany) flying under manual remote control at 50 m a.g.l. (altitude above ground level). The digital camera used for aerial imaging was a Lumix GX7 (Panasonic, Osaka, Japan), mounted on a two-axis gimbal with vibration reducers for stable image capture while in flight. Images were taken at a 16-megapixel resolution of 4592 × 2448 pixels using a 4/3″ sensor and a 20 mm focal length lens for an estimated GSD 0.9419 cm/pixel. These images were taken with a 1/160 second shutter speed and auto-programmed mode for maximum aperture at a rate of every 2 s for the duration of the flight and stored locally on microSD cards for subsequent processing.

The measurements of the color calibration check and the vignetting calibration were taking the same day of the data collection. We used the ColorChecker Passport Photo (X-Rite, Inc. https://www.xrite.com/es/categories/calibration-profiling/colorchecker-passport-photo/), which has a panel of 24 industry standard color reference chips with published values in RGB, as well as the CIELab color space. The photos of this passport were taken with the cameras Olympus OM-D and Lumix GX7 in natural light conditions in a zenithal plane. With the software FIJI (Fiji is Just ImageJ, https://fiji.sc/, https://imagej.nih.gov/ij/), the calibration photos were imported and divided into the separate color channels of red, green, and blue and in the CIELab color space as lightness, a* and b* and then compared with the 24 published reference values of each standard chip with the photos of the passport taken with the different cameras. 

With respect to vignetting calibration, one photo that was taken with the Lumix GX7 at 50 m was divided into the separate RGB and CIELab color space channels. On the R, G, B, hue, a*, and b* single band images, a line was drawn through the center for the X and Y axes in order to extract the cross-image transect. Then, a filter was created using the hue band from the HSI color space in order to select only sunlight soil pixels and applied for R, G, B, a*, and b* to eliminate vegetation and shadowed pixels, and the digital numbers (DN) were extracted from each remaining point along the line in order to observe changes in albedo across the image axes.

NDVI was measured on 28 January 2016 (at the same time as the RGB data) with the GreenSeeker active field sensor (GreenSeeker handheld crop sensor, Trimble, Ukiah, CA, USA), which uses a wavelength range of 650–670 nm and 765–795 nm for red and near-infrared, respectively. Additionally, SPAD chlorophyll meter (Minolta SPAD-502, Spectrum Technologies Inc., Plainfield, IL, USA) measurements were recorded on two different dates (at 3 and 5 weeks after the RGB and NDVI data), once on 18 February 2016 (SPAD vegetative stage, SPAD^V^) and then again on 1 March 2016 (SPAD reproductive stage, SPAD^R^). A total of 4 leaves were measured for each row for a total of 8 measurements per plot to provide a representative average value for each plot. Delayed SPAD sensor timing was due to availability and has been included for sensor technique as well as data capture timing comparisons. Different measurement timing details for the complete study are presented in Figure 3 for added clarity.

### 2.4. Image Processing

For the RGB images captured from the UAV platform, Agisoft PhotoScan Professional software (Agisoft LLC, St. Petersburg, Russia) was employed using a total of 63 overlapping images to produce an accurate image mosaic with at least 80% overlap, and this presented a resolution of 11772 × 4932, as seen in Figure 1. As the aerial images were acquired in clear sky conditions at the same time as the ground RGB images, no cross-calibration radiometric corrections were deemed necessary. The open source image analysis platform FIJI [53] (Fiji is Just ImageJ; http://fiji.sc/Fiji) was used to segment regions of interest for each row for the plots to be cropped in order to produce a single micro image per plot. RGB pictures were subsequently analyzed using a version of Breedpix 0.2 software adapted to JAVA8 and integrated as part of the MaizeScanner, an open-source and open access FIJI plugin that also provides for the implementation of TGI and NGRDI, as well as some specific analyses for maize research-related maize lethal necrosis impact quantification (https://github.com/sckefauver/CIMMYT). 

Within FIJI, images were processed to convert RGB values into indices based on RGB broadband reflectance and also for color quantification from the HSI, CIELab and CIELuv color spaces. The TGI is calculated as the area of a triangle from the matrix determinants after factoring the terms:A = ± 0.5 [(λ1 − λ3) × (R1 − R2) − (λ1 − λ2) × (R1 − R3)](4)where A is the triangular area; λ1, λ2, and λ3 are the center wavelengths for the three image bands; and R1, R2, and R3 are reflectance values for the three image bands, respectively. The order of bands is not important, but the order will affect whether the result is positive or negative. Starting with R1 as R670 (red), R2 as R550 (green), and R3 as R480 (blue) for convenience:TGI = −0.5 [190 × (R670 − R550) − 120 × (R670 − R480)](5)where TGI has units of wavelength × reflectance, so using nm wavelength units or percent reflectance does not affect the value of TGI after the units are converted. We used digital camera bands of red, green, and blue broadband reflectance centered approximately at 670, 550, and 480 nm, respectively, so that λ1, λ2, and λ3 were the centers of the wavebands, and R1–R3 were the waveband reflectance values [54].

We used the NGRDI to analyze the images from the digital camera:NGRDI = (R550 − R670)/(R550 + R670)(6)where R550 and R670 are the reflectance values of the green and red bands of the RGB camera, respectively. The difference between green and red light reflectance differentiates well between plants and soil due to the absorption of chlorophyll at R670, and the sum normalizes for variations in light intensity resulting in a possible range from −1.0 to 1.0, with NGRDI values mostly between −0.2 and 0.5, ranging from soil to healthy vegetation [55].

As described previously, the HSI color space index GA is calculated as the percentage of pixels in the hue range from 60 to 180°, including from yellow to bluish green, while the GGA includes a more restrictive H range from 80 to 180°, excluding yellowish-green tones that might be partially stressed or senescent. Subsequently, the CSI was calculated in agreement with [38,44] as follows: CSI = 100 × (GA − GGA)/GA.(7)

In addition, we developed two new different vegetation indices, modeled after NDVI, such that values of soil fall closer to 0 and vegetation closer to 1. In order to do so, because the a* and the u* image values for green are both negative, those values were placed first but using the complement of a* so that greener vegetation gives a higher value, as would the near-infrared of NDVI. As b* and v* both have more yellowish values with higher values, no inversion was necessary [38,56,57,58]. The normalized difference between a* and b* (NDLab) through the color space CIELab is as follows:NDLab = (((1 − a*) − b*)/((1 − a*) + b*) + 1).(8)

The normalized difference between u* and v* (NDLuv) through the color space CIELuv is as follows: NDLuv = (((1 − u*) − v*)/((1 − u*) + v*) + 1).(9)

By inverting a* and u*, more green vegetation becomes a positive contribution to the index, while more red/brown soil background reduces the index value. Then, dividing by b* and v*, an increase in yellow chlorotic vegetation will reduce the index. The addition of 1 provides for a more balanced equation for positive values for crops from NDLab and NDLuv using CIELab and CIELuv; normalization then limits the index to values between −1 and 1, with most crops between 0 and 1.

### 2.5. Statistical Analysis

Statistical analyses were conducted using the R project for statistical computing [59] in combination with R studio [60]. The maize crop physiological traits were analyzed using ANOVA and Fisher’s Least Significant Difference (LSD) tests (α = 0.05) in order to test the effects of growing conditions on the different traits. The results of the canopy level image averages per picture taken at the ground level were compared with the canopy level whole plot averages of the UAV images (Figure 2) with Pearson correlation coefficients and ANOVA analyses. Pearson correlation coefficients of the different remote sensing indices were additionally compared against grain yield. Multiple regressions were calculated with GY as the dependent variable and the different indices as independent variables using forward stepwise methods with the stepAIC () function of the MASS R package. The figures were also drawn using the R studio software. 

## 3. Results

### 3.1. The Effect of Optimal Condition and Low Managed Nitrogen on grain yield

The range of yield in the LOW treatment was between 1.53 tn/ha and 4.43 tn/ha, while for OP it ranged between 6.68 tn/ha and 12.30 tn/ha; the GYLI range was from 46.88% to 85.22% (Table 1). On the other hand, significant differences in GY between genotypes were observed in this study for the two different conditions (Table A1), but in order to standardize for comparisons between the two treatments, we divided the genotypes into quartiles by yield. Therefore, in Figure 4 the results show the 64 genotypes divided in quartiles as high yield (HY), medium high yield (MHY), medium low yield (MLY), and low yield (LY). The ANOVA for the OP and LOW treatment demonstrated that there were significant differences in GY between all of the quartiles of genotypes. 

In Table 2, the results additionally demonstrate that 44% of the genotypes that belonged in the top HY group grown under OP conditions remained in the HY group in the LOW condition. Similarly, 19% of the genotypes in the OP condition LY group were also in the LOW condition LY group. This suggests that while high yield under both low-N and optimal-N conditions is not completely exclusive, previous breeding efforts have perhaps been more focused on yield in optimal conditions without considering the robust performance of a genotype in other potential growing conditions (i.e., low-N).

### 3.2. The Performance of Remote Sensing Indices and Field Sensors

#### 3.2.1. Color and Vignetting Calibration

The results shown in Figure 5 demonstrate that all the color calibration correlation R^2^ were higher than 0.80, with most falling close to 0.90. With respect to the Lumix GX7, we can see the highest determination coefficient was the green channel, followed by the b*.; for the Olympus Camera the highest was b*, followed by the blue channel. While the L values from the CIELab color space were among the furthest from the 1:1 comparison line with high y-axis intercepts, the associated a* and b* linear correlations were among the closest to a 1:1 ratio with intercepts close to 0. 

Figure 6 shows that the vignetting effects observed for the both color spaces with respect to the y- and x-axes were minimally present for the RGB color spaces and reduced for a* and b* in CIELab.

While the calibration check for the RGB images taken from both cameras demonstrated high correlations (Figure 5), the result of applying the calibration coefficients to the data resulted in both cases in lower correlations between the vegetation indices at different scales, as well as lower correlations between the RGB indices with GY (data not shown). Moreover, the results demonstrated low presence of vignetting effects, with reduced vignetting in the luminescence-controlled CIELab and CIELuv color spaces of particular interest for this study (Figure 6).

#### 3.2.2. The Performance of Remote Sensing Indices and Field Sensors Assessing Grain Yield

No significant differences were found between quartile groups for any of the RGB indices from the aerial or ground level, as seen for GGA at ground level. The correlations were calculated for GY with both levels of RGB indices at LOW (Table 3). In the case of UAV RGB VIs, GGA was best correlated with GY followed by CSI, followed by saturation and GA. For ground RGB VIs the closest correlations were observed with hue, GGA (values between 0.248 and 0.685), and NDLab (values between 0.3953 and 0.8290). The rest of the RGB VIs were somewhat weaker with respect to the GY, but many were still significant. Additionally, the field sensors presented some close correlations with GY (Table 3), as well as significant differences between genotypes when grouped by quartile (data not shown), the strongest was SPAD^V^ (taken in the vegetative stage closer to the RGB VIs), which only indicated differences between HY and LY, followed by SPAD^V^ and finally NDVI, both of which were recorded in the vegetative stage (see Figure 3). SPAD measurements exhibited the highest correlations (Table 3) with respect to all indices with SPAD^V^ being the highest followed by SPAD^R^ (taken in the reproductive stage much later than the other measurements for temporal comparisons only).

The correlation coefficients between the hue, u*, GA, and GGA remote sensing indices evaluated at ground level versus the same indices measured from the UAV were quite strong (Table 4). In addition, most of these indices showed slopes close to 1:1 and correlations reaching r = 0.766. In contrast, the relationships reported for the remaining RGB indices, such as intensity, lightness, TGI, and NGRDI were lower. With regards to ANOVA, the results showed that there were statistically significant differences between all the RGB indices at ground level with the aerial observation level, except for saturation from the HSI color space.

### 3.3. Agronomic Parameters and Their Effect on Yield

Finally, for agronomic data parameters, the correlations with GY (Table 5), the results showed that all the agronomic data indicators performed differently between OP and LOW conditions. For LOW conditions, the indices that were better correlated with GY were ASI and AD. The other two indices (PH and SEN) showed very low correlations with GY. In OP conditions, the agronomic parameters showed very low correlations with GY.

### 3.4. Multivariate Models

Figure 7 shows the correlations of the most relevant agronomic parameters and indices with GY; it was considered that these could be complimentary in multivariate models because ASI and AD show negative correlations with respect to GY, whereas the other indices present positive correlations with GY. As such, in Table 6 we present the stepwise multivariate linear models for explaining grain yield using different selections of non-destructive UAV RGB VIs and ground RGB VIs at additional field sensor and agronomic data as indicated using both forward and backward stepwise selection techniques with a standard Akaike information criterion (AIC) selection criterion. We also present the determination coefficients (R^2^) and the residual standard error (RSE). All three models presented were found to be significant at the P < 0.001 level. Where noted, (*) indicates simplified formulas, meaning that in using stepwise selection these formulas were considered to have an excess of nonsignificant parameters and were reduced accordingly (only significant parameters with the strongest individual correlation to yield were selected in the case of auto-correlation detected between two multivariate parameters).

In the combination of agronomic data with additional field sensors, such as SPAD^v^, NDVI, and ASI, 61% of the yield could be explained. The multivariate stepwise models for explaining yield variations regarding agronomic data plus ground RGB VIs level was 63%, but in Table 5, we show the simplified formula explaining 58% of variance. With respect to combining agronomic data with UAV RGB VIs, the result was similar at 62%, but we show the simplified formula with 60% of the yield that may be explained by mostly agronomic data and combined with other indices, such as NDLab, NDLuv, and Saturation. Combining the agronomic data, field sensors and RGB VIs provided little improvement in the multivariate model explaining the yield, whereas, in comparison, more parsimonious models combining only AD, ASI, and only either the NDLab or NDLuv RGB indices still explained over 50% of the variation in yield (data not shown). 

## 4. Discussion

### 4.1. The Effect of Managed Low Nitrogen on Grain Yield

Nitrogen (N), after water, is the single most important input for maize production. It plays a major role in establishing optimal photosynthetic capacity during key growth stages for crops to achieve high yields [18,61]. N deficiency reduces leaf chlorophyll content, soluble protein content, photosynthetic rate, and related enzyme activities of the maize plant during grain filling [62,63,64]. For that reason, the GY of all the 64 genotypes was assumed to have been strongly affected by the lack of nitrogen in LOW conditions. There were significant differences noted in GY by genotype, but in Table A1 we can see that OP treatment presented many more different groups, and because of that, the genotypes were divided into quartiles for the sake of comparisons across growing conditions. (Figure 4). These four groups showed differences between each other with respect to grain yield. Nitrogen is especially plentiful in leaves, mainly in photosynthetic enzymes, where it may account for up to 4% of the dry weight. Because N uptake, biomass production, and grain yield are strongly correlated, the N requirement of a maize crop has even been directly related to grain yield; it has been estimated that 187 kg/ha N is required to produce 9.5 t/ha yield, 98 kg/ha is required for 5.0 t/ha, and 40 kg/ha is required for 2.0 t/ha [17]. Following these guidelines, the concentration for the optimal condition was around 200 kg/ha and that for the low managed nitrogen condition was around 40 kg/ha, thus indicating that many of the genotypes tested here have already been somewhat adapted to the managed low nitrogen conditions; however, those that showed the best and most consistent adaptation were the genotypes of the HY quartile, some of which appeared in HY in both LOW and OP (Table 2). In some cases, it has been reported that the genotypes selected under LOW fertilization input are not truly adapted to N-rich soils [65]; however, it has been suggested that when the plant material performs relatively well under low-N input conditions, it should be selected under N deficiency conditions for which yield reduction does not exceed 35–40% [66]. Here, in comparison with the same panel of maize genotypes grown under optimal conditions, 44% of the genotypes that were in the highest yield producing quartile under OP conditions remained in the highest quartile when grown under LOW conditions, further suggesting that low N productivity is not necessarily exclusive of high productivity in OP conditions. The GYLI results also show that there was a large amount of genotypic variability present (Table A1), again suggesting that the genotypes selected for this study behaved quite differently physiologically under the two nitrogen conditions.

### 4.2. Effect of Managed Low Nitrogen on Agronomic Parameters 

A higher reduction in maize yield under stress environments can often be partly explained by the wider date range of ASI under stress, as ASI typically has a high negative correlation with GY under stress conditions [67,68]. In Figure 7, we demonstrated that AD and ASI exhibit negative significant relation in low-N conditions correlations when the plants were under stress, as these parameters decreased with increasing yield. Genotypes with a short ASI have been suggested to possess greater efficiency in biomass partitioning to ear and tassels at flowering than those with a long ASI [69]. On the other hand, the correlation results in OP conditions do not indicate any relationship between GY and ASI or AD (Table 5). Similar studies showed that high GY under a range of stress intensities is associated with a short ASI and earlier flowering dates, increased plant and ear height, increased number of ears per plant, and delayed leaf senescence [69,70].

Besides ASI, traits related to high photosynthetic capacity (e.g., chlorophyll content) and plant water content (e.g., stomatal conductance) have often been reported to contribute to higher GY under drought stress [71]. When maize flowers under drought, there is a delay in silking, and the period between male and female flowering increases giving rise to ASI. In this study, however, drought stress was minimal due to the adequate rainfall recorded during the study field season (700 mm), and also further supported by the lack of correlation between ASI and AD with GY in the OP conditions (Table 5). In generally optimal agronomic conditions, these phenological characteristics are not always good estimators of yield. In contrast, the rest of the agronomic parameters here showed similar relation with GY variability. These results suggest that this technology could be applied in an adapted way to water stress studies, even though it was not the specific aim of this research.

### 4.3. Remote Sensing Indices and Field Sensors

#### 4.3.1. Color and Vignetting Calibration

Figure 5a–c,g–i show the correlations between the R, G, and B values from of each digital camera with respect to the standard values for R, G, and B from the X-Rite ColorChecker Passport. Some lamination effects can be observed as values from the camera appearing higher than the standard, though still with good correlation. However, in Figure 5e,f,k,l of the a* and b* images, the light effects are less pronounced, with a ratio closer to 1:1 due to the CIELab color space separation of lightness from color hue values [72]. Regarding the color calibration, the results show that the determination coefficients between the values of the ColorChecker passport and the values for each camera were high (most near R^2^ ≈ 0.90), suggesting that the photos were not in need of a separate color calibration [73]. Indeed, application of the color calibration coefficients did not result in any improvements in RGB vegetation index performance with regards to grain yield or yield loss estimation (data not shown). 

With respect to vignetting effects, in Figure 6a,b, the graphics of three RGB channels present a small sad smile along the axis, showing some brightness attenuation away from the image center [74,75]. Nevertheless, this did not represent any significant differences between the DN values of extremes of each side respect to the DN of center (data not shown). On the other hand, Figure 6c,d present minimal variation along the axis. This was expected as a* and b* color spaces are independent of the image lightness and thus absent of vignetting effects. Thus, while the RGB VIs are passive sensors and dependent on ambient light conditions, the use of the alternate color spaces, such as CIELab, provides for inherent lightness correction and enables their use in variable conditions similar to active sensors, such as the NDVI GreenSeeker.

#### 4.3.2. Performance of RGB VIs and Additional Field Sensors

The ground RGB VIs hue, GGA, and NDLab demonstrated the best correlations with GY, outperforming other ground RGB VIs (Table 3). GA and GGA quantify the portion of green pixels to the total pixels of the image and is a reliable estimation of vegetation cover [76]. The values of GA from both observation levels (field and aerial) were consistently below 60%. The ground and aerial measurements were taken at the same time on the same day, variation in environmental variables, such as light intensity and brightness can be assumed to be negligible. Thus, the main differences must be due to the resolution of the images (Figure 2); nevertheless, advances in digital photography allow for sufficiently high resolution for low-altitude aerial imaging to be a viable and economical monitoring tool for agriculture [77]. In this sense, correlations with GY by indices derived from aerial images were generally only slightly weaker than indices measured at ground level, most likely demonstrating a trade-off between them. Some of the RGB indices, such as NDLab, GGA, and GA produced coefficients of correlation higher than r = 0.75 when comparing the same indices measured from the ground level and from the aerial platform (Table 4). This is despite different acquisition/imaging techniques (full plot for UAV RGB VIs vs. one image per plot at higher resolution covering only a portion of the plot for the ground RGB VIs). On the other hand, none of the UAV RGB VIs and ground RGB VIs showed significant differences between adjacent quartiles. This may be best explained considering that the data for our study were collected at an early phenological (vegetative) stage, when the plants were not yet at full canopy cover, and they did not yet show the full range of symptoms of N deficiency, as may be observed in the reproductive stage (Figure 3). N deficiency can reduce plant growth rates, but also other later factors that affect GY, including leaf chlorophyll content, soluble protein content, photosynthetic rate, and related enzyme activities of the maize plant during grain filling in the reproductive stage [78,79,80], which may be a more optimal timing of remote sensing observations when phenotyping low N.

NDVI has been used with satisfactory results in many predictive models of yield in multiple crops, including wheat, barley, and maize at the field level [54,81], even at regional or state levels using field, airborne, and satellite imagery [82,83,84]. Regarding NDVI, the values clearly highlight (Figure 7) that the variability was low, with more than 90% of values being in the range of 0.55–0.80. These results support the previously reported saturation of the index, such that increasing leaf area does not involve a parallel increase in NDVI values [83,85,86]. Furthermore, other authors have noted that the optimal stages for measuring NDVI vary depending on the germplasm and environment [87,88]. Better performance of NDVI usually occurs at earlier or later growth stages, depending on the crop and symptoms, because at maximum vegetation cover, NDVI values often saturate, and thus, correlation with GY decreases [38]. Furthermore, other studies comparing the performance of RGB VIs with frequent data acquisitions throughout the crop growing cycle and NDVI indicate that RGB VIs respond with higher correlations with GY earlier than NDVI [57].

SPAD is used to measure relative chlorophyll content in plant leaves, and it has been effectively used to diagnose N status and predict GY potential in maize [89]. Leaf level chlorophyll meters provide a convenient and reliable way to estimate leaf N content during vegetative growth [90] and over a large time range even after anthesis [91]. With specific mention regarding the two different measurement dates, SPAD^V^ and SPAD^R^ (Table A2), there are some notable differences related, in this case, to the date of measurement, having been one of the few sensors available to the field crews on-site to do multiple measurements. In the first measurement, SPAD^V^, the results showed significant differences between the HY, MHY, and MLY groups in comparison with the LY group. These are interpreted as symptoms of lack of nitrogen. With regards to the second measurement, SPAD^R^, it was possible to identify the differentiation of HY groups in comparison with MHY and MLY and, additionally, these three from the LY group. This is interpreted as a decline in relative chlorophyll content of the leaves measured between the two SPAD measurements. This is because when crops were younger at SPAD^V^, when still developing roots and leaves, they may behave as sink organs for the assimilation of N and the synthesis of amino acids originating from N uptake before flowering [92]. After flowering, at SPAD^R^, the N accumulates in the vegetative parts of the plant and is remobilized and translocated to the grain [93]. In most crop species, a substantial amount of N is absorbed after flowering to contribute to grain protein deposition [89] in V8–R1. Similarly, Teal et al. [94] also reported a strong association between grain yield and NDVI between the V6 and V8–R1 growth stages of maize, again between the timing of SPAD^V^ and SPAD^R^, but after the remote sensing observations were recorded. Finally, the increased performance of SPAD^V^ compared with SPAD^R^, closer to the RGB and NDVI data acquisition date, was promising in that earlier stage of image data capture for field phenotyping, which may reduce crop breeding costs with earlier variety selection and increased crop cycles per year.

### 4.4. Multivariate Model Assessment

The vegetation indices derived from conventional digital RGB images have been proposed as means of estimating green biomass and GY in maize and other cereals under stress conditions [95], and in other studies in wheat grown under different stress conditions [38,43,96]. The multivariate regression models revealed the most appropriate parameters for field phenotyping towards improving GY in managed low-nitrogen conditions. Using all the UAV RGB VIs and ground UAV VIS, the multivariate models explained GY at around R^2^ = 0.30 (data not shown). That could be a result of the fact that the data capture of RGB VIs were taken earlier than the SPAD^V^ and SPAD^R^, and at later growth stages, the plants may have presented more symptoms related to a lack of nitrogen. However, all of the regression models with a R^2^ higher than 0.50 included some of the agronomic data as independent variables. Additionally, GY estimation was similar in the cases of agronomic data combined with field sensors with respect to agronomic data combined with UAV RGB VIs and ground RGB VIs (all having approximate R^2^ values of 0.60). With respect to the RSE in these three different multivariable models, they adapt to a 50% coefficient of determination. As such, our study suggests that these phenological traits still provide useful information related to grain yield in abiotic stress conditions, but that they may be potentially supplemented by UAV RGB VIs and ground RGB VIs phenotyping platforms. Still, RGB image analyses were able to improve over agronomic data alone, increasing the R^2^ values to explain more than half of the variance in the yield, suggesting that they are complimentary in the information that they are able to provide. Furthermore, the UAV RGB VIs in this study were acquired quite early in the growing season, which may help to provide for faster selection of varieties, thus reducing costs and increasing the number of crop cycles per year.

Additionally, RGB VIs may provide considerable saving with regards to field equipment and human time, considering that RGB data capture and processing of 200 plots took approximately 10 min in the field (counting flight and preparations), 20 min to mosaic (unattended on the computer), and half an hour to extract and process the data (semi-automatic), totaling 60 min, excluding drone preparation prior to flying. In the case of RGB image data capture and processing, the field portion would be approximately double, while the computer processing would be about half, totaling about the same amount of time in the case of 200 plots. With respect to time costs while implementing the use of SPAD or field-based GreenSeeker NDVI, the estimated time for measurement would be over five times greater (e.g., 2 min per plot × 200 plots = 400 minutes). Moreover, NDVI evaluation from the ground may not be easy to implement when plants are reaching the reproductive stage. Furthermore, with larger phenotyping trials, the time savings of the UAV RGB VIs would represent even greater time savings while retaining the same data quality; up to 1000 plots may approximately double the amount of time needed to process the data, while the field sensors would increase linearly and take five times as long, representing over a 10-fold time difference at larger study scales. Thus, the implementation of higher throughput UAV RGB VIs may make the most sense in combination with some of the quicker traditional agronomic measurements and can also result in substantial time cost savings when applied in large platform breeding programs. 

In a recent study by Gracia-Romero et al. in 2017 [97], the effectiveness of UAVs for canopy level remote sensing for plant phenotyping of maize was similarly demonstrated under different phosphorus nutrient conditions and the results presented therein suggested that the RGB indices were the best option at early growth stages. In the case of low P, however, an equation using GA and u* were the best indicators of GY (R^2^ = 0.82). Even though applied to a different crop in that case, reflectance in the near-infrared (NIR) and blue regions was found to predict early season P stress between growth stages V6 and V8, much earlier than suggested for N deficiencies. With respect to plant N concentrations, the best correlations have been found using reflectance in the red and green regions of the spectrum, while grain yield was best estimated using reflectance in the NIR region, with the wavelengths of importance changing with growth stage (V14-R1) [98,99]. Furthermore, Ma et al. [100] showed that canopy light reflectance is strongly correlated with field “greenness” (similar to the GA and GGA used in this study) at almost all growth stages, with field greenness in that case being a product of plant leaf area and leaf greenness measured with a chlorophyll meter.

## 5. Conclusions

Modern phenotyping technologies may help in improving much-needed maize GY in low-N conditions, and the current range of variability in performance as indicated by the observed GYLI values suggests that low-N and optimal-N performance need not be considered mutually exclusive. For HTPP, RGB sensors can be considered to be functional technology with an advanced technology readiness level (TRL) from the ground or a UAV platform, but, similar to the current standard field sensors SPAD and NDVI, the data capture for RGB VIs must be planned accordingly in order to optimize their benefits in support of plant breeding. Several different RGB image-based vegetation indices, including the NDLab and NDluv indices new to this study, demonstrated similar correlations with GY and contributions to multivariate model GY estimates when compared with standard NDVI and SPAD field phenotyping sensors. This study presents possible uses of RGB color image analyses from the ground or from UAVs, with potential benefits compared with currently used field sensors, especially regarding time costs when applied to larger breeding platforms, here demonstrated in application to low-N phenotyping in maize.

## Figures and Tables

**Figure 1 sensors-19-01815-f001:**
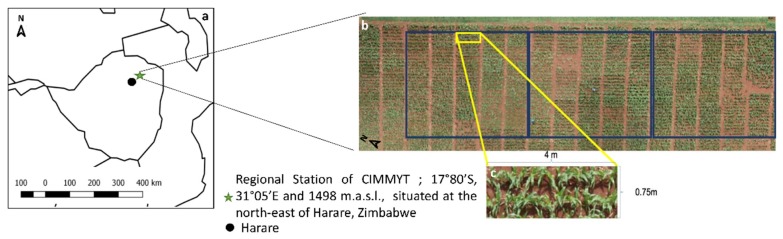
(**a**) Position of the regional station of International Center for Maize and Wheat Improvement (CIMMYT) in Harare, Zimbabwe. (**b**) Red–green–blue (RGB) aerial orthomosaic of the 192 plots, with 64 genotypes and 3 replicas (three blue box) per each one, under low managed nitrogen (LOW). (**c**) A plot with specific details of length and width.

**Figure 2 sensors-19-01815-f002:**
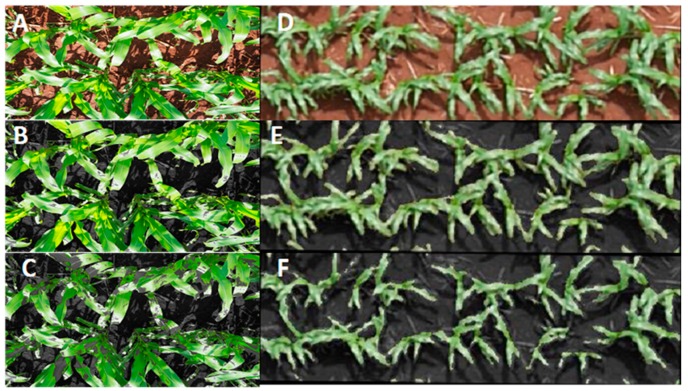
Examples of the differences in resolution between the images of maize taken at ground level (Ground Sample Distance 0.03 cm/pixel) and aerial level (Ground Sample Distance 0.9419 cm/pixel) in LOW. (**A**) Maize ground image from 80 cm or canopy level image averages. (**B**) Maize ground image from 80 cm showing green area (GA). (**C**) Maize ground image from 80 cm with greener green area (GGA). (**D**) Maize aerial image from 50 m or canopy level whole plot averages. (**E**) Maize aerial image from 50 m showing GA. (**F**) Maize aerial image from 50 m showing GGA.

**Figure 3 sensors-19-01815-f003:**
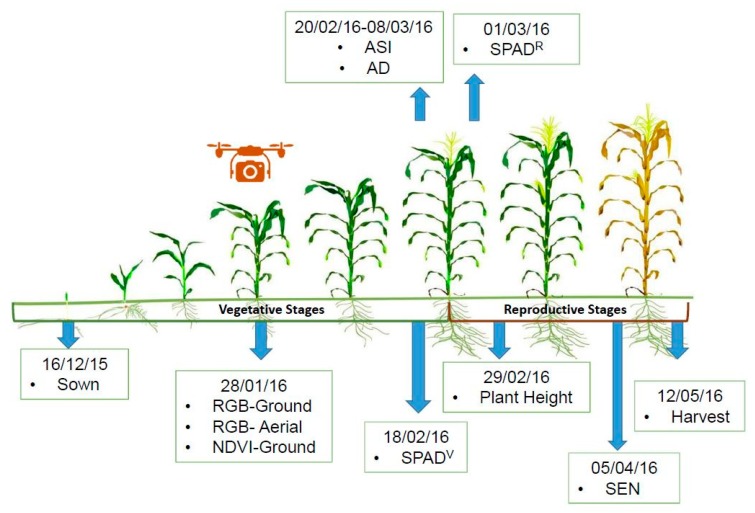
Field phenotyping, field imaging, and unmanned aerial vehicle (UAV) aerial image data capture chronogram for the controlled low-N field trial, showing dates for the measurement of all parameters at vegetative and reproductive stages, including red–green–blue (RGB), (high throughput plant phenotyping (HTPP) imaging, plant height (PH), canopy senescence (SEN), anthesis data (AD), and anthesis silking interval (ASI).

**Figure 4 sensors-19-01815-f004:**
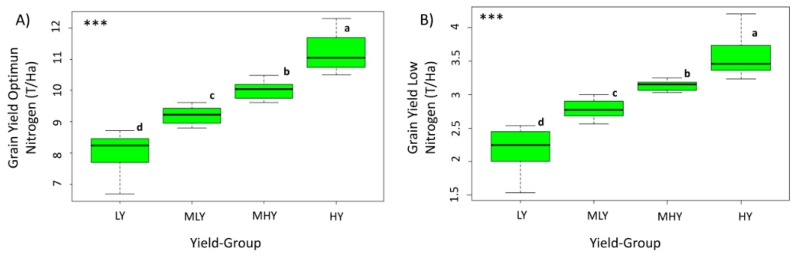
Box plot of grain yield for the set of 64 genotypes divided into four quartiles: low yield (LY), medium low yield (MLY), medium high yield (MHY), and high yield (HY) under OP (**A**) and LOW (**B**) conditions. The bottom and top of the box are lower and upper quartiles, respectively. The band near the middle is the median value across each group, and the bars are the standard deviation. Letters are significantly different according to Fisher Least Significant Difference (LSD) multiple range test (*P* < 0.01).

**Figure 5 sensors-19-01815-f005:**
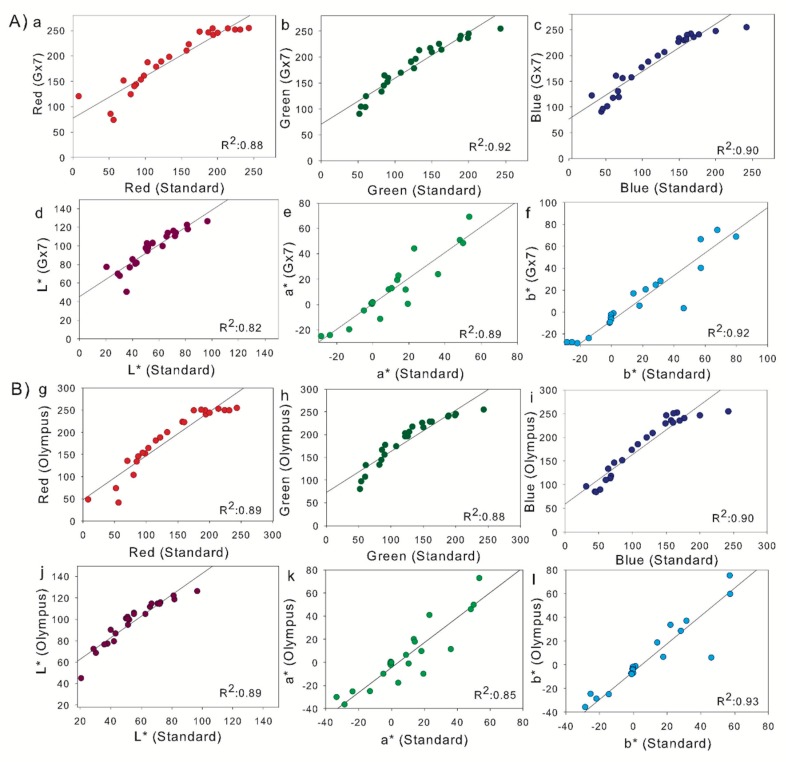
Color calibration through ColorChecker Passport. (**A**) The determination coefficients between the photo taken with Lumix GX7 camera for each channel red, green, and blue (**a**–**c**) and Commission Internationale de I´Edairage (CIE)Lab color space (**d**–**f**) (n = 24). (**B**) The determination coefficients between the photo taken with the Olympus OM-D camera for each channel red, green, and blue (**g**–**i**) and CIELab color space (**j**–**l**) (n = 24).

**Figure 6 sensors-19-01815-f006:**
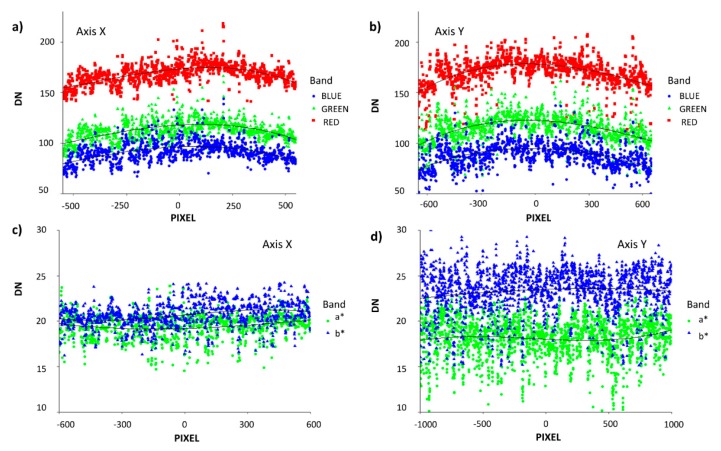
Vignetting effects observed in the RGB and CIELab color spaces from one example photo taken with Lumix GX7 from the UAV at 50 m, comparing x-axis and y-axis transect pixel digital number (DN) values. Hue-based color filters were used to identify only bare soil pixels along each transect. RGB channels (**a**,**b**). CIELab color space without lightness (**c**,**d**).

**Figure 7 sensors-19-01815-f007:**
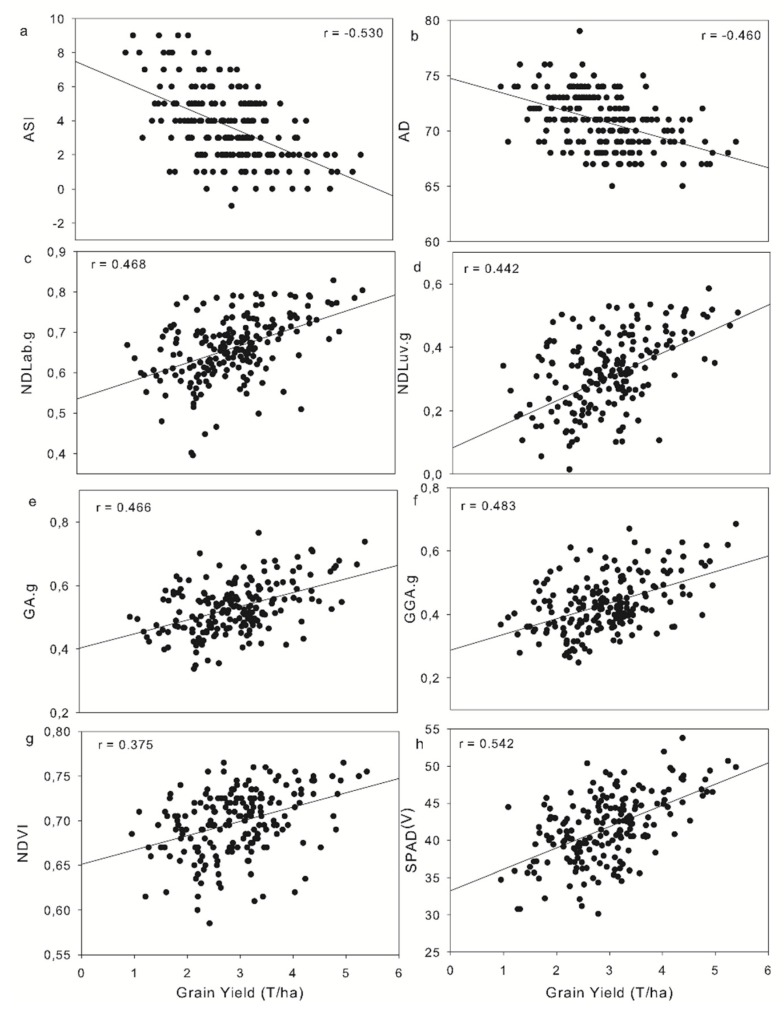
Correlation between GY in LOW and (**a**,**b**) traditional field plant physiology measurements ASI and AD; ground RGB VIs (**c**) NDLab; (**d**) NDLuv; (**e**) GA.g (green area ground); (**f**) GGA.g (greener green area ground); (**g**) plant vigor using GreenSeeker NDVI; and (**h**) relative leaf chlorophyll content in the vegetative stage (SPAD^V^). Level of significance: ***, *P* < 0.001. (n = 192).

**Table 1 sensors-19-01815-t001:** Minimum, maximum, and average of grain yield (GY) and percentage of Nitrogen (N) of the two different treatments: LOW and optimum nitrogen (OP). Minimum, maximum, and average of grain yield loss index (GYLI).

	Minimum	Maximum	Average	N (%)
GY (Mg/ha) at LOW	1.53	4.43	2.93 ± 0.58	25–35
GY (Mg/ha) at OP	6.68	12.30	9.62 ± 1.24	100
GYLI (%)	46.88	85.22	69.01 ± 7.48	

**Table 2 sensors-19-01815-t002:** Maize genotypes that were in both, the HY and LY groups with different applications of nitrogen: LOW and OP, with their GY.

	LOW		OP	
Genotype	GY(Mg/ha)	Yield Group	GY(Mg/ha)	Yield Group
CZH128	3.35	HY	10.93	HY
CZH15024	3.50	HY	11.13	HY
CZH15028	3.88	HY	10.60	HY
CZH15045	3.38	HY	10.13	HY
CZH15057	3.82	HY	11.40	HY
11C4393	3.37	HY	10.97	HY
LOCAL CHECK2	2.44	HY	10.59	HY
CZH15027	2.32	LY	8.44	LY
PHB30G19	2.28	LY	8.46	LY
MRI 634	2.29	LY	7.97	LY

**Table 3 sensors-19-01815-t003:** Grain yield correlations in LOW with all proximal remote sensing variables from the RGB images taken from the UAV aerial platform, RGB images from the ground, and leaf chlorophyll content (SPAD) and normalized difference vegetation index (NDVI) field sensors. These indices are defined in Section 1 and Section 2. Levels of significance: *, *P* < 0.05; ***, *P* < 0.001. GGA; GA; NDLab, normalized difference between a* and b*; NDLuv, the normalized difference between u* and v*; CSI, crop senescence index; TGI, triangular greenness index; and NGRDI, normalized green–red difference index.

GY								
UAV RGB VIs	r	*P*	Ground RGB VIs	r	*P*	Additional Field Sensors	r	*P*
GGA	0.445	***	GGA	0.483	***	SPAD^V^ (18/02/16)	0.542	***
GY	0.407	***	GA	0.466	***	SPAD^R^ (01/03/16)	0.506	***
Hue	0.381	***	Hue	0.485	***	NDVI	0.375	***
Intensity	−0.305	***	Intensity	0.095				
Saturation	−0.427	***	Saturation	−0.227	*			
Lightness	−0.291	***	Lightness	0.144	*			
a*	−0.36	***	a*	−0.383	***			
b*	−0.397	***	b*	−0.089				
u*	−0.383	***	u*	−0.449	***			
v*	−0.297	***	v*	0.014				
NDLab	0.359	***	NDLab	0.468	***			
NDLuv	−0.378	***	NDLuv	0.442	***			
CSI	−0.428	***	CSI	−0.321	***			
TGI	0.229	*	TGI	−0.043				
NGRDI	0.406	***	NGRDI	−0.027				

**Table 4 sensors-19-01815-t004:** Correlation coefficients between the remote sensing UAV RGB VIs and ground RGB VIs. These indices are defined in Section 1 and further detailed in Section 2. Levels of significance: *, *P* < 0.05; **, *P* < 0.01, ***; *P* < 0.001; ns, not significant. GGA, GA, NDLab, NDLuv, CSI, TGI, and NGRDI.

	R	*P*	ANOVA
GGA	0.758	***	***
GA	0.766	***	***
Hue	0.731	***	***
Intensity	−0.062	ns	***
Saturation	0.509	***	ns
Lightness	−0.039	ns	***
a*	0.617	***	***
b*	0.424	***	***
u*	0.723	***	***
v*	0.33	***	***
NDLab	0.781	***	***
NDLuv	−0.676	***	***
CSI	0.457	***	***
TGI	−0.163	*	*
NGRDI	−0.223	*	**

**Table 5 sensors-19-01815-t005:** Grain yield correlations with different indices of agronomic data, such as PH, SEN, AD, and ASI. Correlations were studied across plots in LOW and OP conditions. Levels of significance: *, *P* < 0.05: **, *P* < 0.01; ***, *P* < 0.001; ns, not significant.

GY				
Agronomic Data	LOW		OP	
	r		r	
PH	0.191	**	0.131	ns
SEN	−0.213	**	NA	ns
AD	−0.46	***	0.272	**
ASI	−0.53	***	0.161	*

**Table 6 sensors-19-01815-t006:** Multilinear regressions (stepwise) of GY in LOW as the dependent variable comparing the different categories of remote sensing traits: UAV and ground RGB VIs (these indices are defined in Section 1), agronomic data such as ASI, AD, SEN, and PH, and NDVI and SPAD. R^2^, determination coefficient; RSE, Residual Standard Error. Level of significance: ***, *P* < 0.001. (*) simplified formulas.

Parameters	Stepwise Equations	R^2^	RSE	*P*
Agronomic Data + Field sensors	GY = − AD*0.28 + SPAD^V^*0.03 + SPAD^R^*0.02 − ASI*0.78 + 5.97	0.61	0.539	***
Agronomic Data + Ground RGB VIs (*)	GY = − ASI*0.189 − AD*0.128 − SEN*0.237 + PH*0.01 + b*0.11 − v*0.064 + NDLab*15.20 − NDLuv*6.99 + 3.36	0.588	0.556	***
Agronomic Data + UAV RGB VIs (*)	GY = − ASI*0.20 − SEN*0.26 − AD*0.13 + PH*0.01 − Saturation*84.97 − u*1.37 + v*1.61 + TGI*0.02 + NDLuv*3.95 + 31.8	0.604	0.546	***

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
