# Peer review of "Evaluating Maize Genotype Performance under Low Nitrogen Conditions Using RGB UAV Phenotyping Techniques"

_sensors, 2019, doi:10.3390/s19081815_

Reviewer 1 Report

This topic is interesting and might attract broad readers. Moreover, the paper is well organized and well written. However, there are still some points should be improved.

(1)     This study lack measuring the plant physiological parameters, which could be used to indicate the rationality of the used spectral index. If you did not measure these parameters, you can cite some references to explain these spectral indexes.

(2)     I do not agree the multivariate modes to predicate the grain yield, since the independent variables have overlapped meaning on the maize growth. Moreover, these models are statistical models, which can not be further used in other studies. 

Author Response

We appreciate the reviewer’s comments and have worked to improve the description details of the Methods, the clarity of the Results and have also refined our Conclusions with the aim of receiving improved marks above. Specifically, more details have been added with regards to the meaning and interpretation of the RGB vegetation indices in the Methods and the Results, and the Conclusions have been almost entirely reframed, as detailed below in our specific comments regarding (1) and (2).

(1)        This study lack measuring the plant physiological parameters, which could be used to indicate the rationality of the used spectral index. If you did not measure these parameters, you can cite some references to explain these spectral indexes.

This focus of this manuscript is on the use of RGB commercial digital cameras for the rapid quantification of crop plant traits through the use of RGB vegetation indices (RGB-VIs) like Hue, a*, b*, GA, GGA, NGRDI, and the new NDLab and NDLuv. These are examined for their potential as affordable HTPP tools through comparison to two standard HTPP field tool, namely the SPAD chlorophyll content and Greenseeker NDVI sensors, as well as for their capacity to provide meaningful phenotypic data regarding different maize genotypes under low and optimal N conditions. Some of the RGB-VIs are from previous studies (Hue, a-, b* GA, GGA, NGRDI) where their results have been compared to plant physiological parameters and have been published as phenotypic tools recently. Both the SPAD and NDVI sensors are highly published and have been cited in support of the basis as indexes related to real plant physiological parameters within this manuscript. The two newly presented RGB-VIs, NDLab and NDLuv have been afforded some additional supporting text in these revisions, thus providing more rationale for their development and conceptualization as to their meaning in terms of relationships to the SPAD (chlorophyll/pigments) and NDVI (fractional vegetation cover/biomass) sensors as follows. When introducing the alternate color space concepts and potential uses, we have expanded the Introduction with more detail on lines 140-143:

“Thus, the CIE-Lab and CIE-Luv color spaces offer the ability to simultaneously contrast green vegetation quantity with both the reddish/brown soil background (fractional vegetation cover or plant growth) and yellowing caused by chlorosis (loss of foliar chlorophyll), both common symptoms of nitrogen deficiency.”

And also, later on in the Methods with an expansion on the author’s expert interpretation of the resulting values from the NDLab and NDLuv RGB-VIs in Lines 341-345:

“By inverting a* and u*, more green vegetation becomes a positive contribution to the index while more red/brown soil background reduces the index value. Then, division by b* and v* an increase in yellow chlorotic vegetation will reduce the index. The addition of 1 to the denominator provides for a more balance equation given common values for crops in CIE-Lab and CIE-Luv and normalization then limits the index to values between -1 and 1.”

We have added to the paper some agronomic data relevant to the study in order to demonstrate that these tools are not the full solution to the understanding of plant physiological responses to such a large change in N fertilizer; however, they may be considered complimentary to some fairly rapidly assessed agronomic data and may provide meaningful data and results towards plant phenotyping at a similar level of benefit as other currently used sensors.

(2)        I  do not agree the multivariate modes to predicate the grain yield, since  the independent variables have overlapped meaning on the maize growth.  Moreover, these models are statistical models, which can not be further  used in other studies.

This was an error of the authors and has been corrected, we changed the “predicted” to “estimate or explain” and we furthermore removed the last two equations from Table 6, as they were only shown in order to demonstrate that the combined field sensors, VI-RGB and agronomic data model is very similar in its capacity to estimate or explain GY compared to the first four multivariate stepwise models. Thus, we only present the most relevant comparisons of agronomic data combined with the field sensors, agronomic data combined with RGB-VIs at the ground level, and agronomic data combined with RGB-VIs at aerial level. Below are the relevant texts or changes in text:

Lines 492-507:

Table 6. Multilinear regressions (stepwise) of Grain Yield (GY) in LOW as the dependent variable comparing the different categories of remote sensing traits: RGB ground and aerial level (these indices are defined in the Introduction), agronomic data like ASI (Anthesis Silking Interval), AD (Anthesis Data), SEN (Canopy Senescence) and PH (Plant Height) NDVI (Normalized Difference Vegetation Index) and SPAD (relative leaf chlorophyll content). R2, determination coefficient; RSE, Residual Standard Error. Level of significance: ***, P<0.001. (*) simplified formulas.

Parameters

Stepwise Equations

R2

RSE

P

Agronomic Data + Field sensors

GY=-AD*0.28+SPADV*0.03+SPADR*0.02-ASI*0.78+5.97

0.61

0.539

***

Agronomic Data + RGB ground (*)

GY=-ASI*0.189-AD*0.128-SEN*0.237+PH*0.01+b.g*0.11-v.g*0.064+NDLab.g*15.20-NDLuv.g*6.99+3.36

0.588

0.556

***

Agronomic Data + RGB aerial (*)

GY=-ASI*0.20-SEN*0.26-AD*0.13+PH*0.01-sat.a*84.97-u.a*1.37+   v.a*1.61+tgi.a*0.02+NDLuv.a*3.95+31.8

0.604

0.546

***

In the combination of agronomic data with additional field sensors, such as SPADv, NDVI and ASI, 61% of the yield could be explained. The multivariate stepwise models for yield estimation regarding agronomic data plus VIs at ground level was 63%, but in Table 5 we show the simplified formula estimating 58% of variance. With respect to combining agronomic data with RGB-VIs at aerial level, the result was similar at 62%, but we show the simplified formula with 60% of the yield that may be explained by mostly agronomic data and combined with other indices like NDLab, NDLuv, Saturation. Combining the agronomic data, field sensors and RGB indexes provided little improvement in the multivariate model estimations of yield, while, in comparison, more parsimonious models combining only AD, ASI and only either the NDLab or NDLuv RGB indexes still explained over 50% of the variation in yield (data not shown). “

Lines 631-642 “The vegetation indices derived from conventional digital RGB images have been proposed as means of estimating green biomass and GY in maize and other cereals under stress conditions [92], and in other studies in wheat grown under different stress conditions [38,43,93]. The multivariate regression models revealed the most appropriate parameters for field phenotyping towards improving GY in managed low nitrogen conditions. According to RGB-VIs at both observation levels, the modeled GY estimation was around R2=0.35. That could be a result of the fact that the data capture of RGB indices were taken earlier than the SPADV and SPADR, and at later growth stages the plants may have presented more symptoms related to a lack of nitrogen. However, all of the regression models with a R2 higher than 0.50 included some of the agronomic indicators as estimate parameters. Additionally, GY estimation was similar in the cases of agronomic indicators combined with field sensors with respect to agronomic indicators combined with RGB from either the UAV or ground level (all having approximate R2 values of 0.60).”

We thank the reviewer for their highly constructive criticisms and have worked diligently to provide detailed explanations and improvements in the manuscript following their suggestions. In some cases, we felt that the changes requested by the other reviewer and the previously presented text had already nearly answered their requests, and have made small changes in a few locations, but larger changes in others. We are certainly amenable to making further adjustments to the manuscript should there be any other points that require further detail or clarification.

Reviewer 2 Report

Dear Authors, 

Please find below my comments and suggested edits. A major concern is that it is not clear to me how your VI values were extracted from your UAV versus ground measurements to perform correlation analysis - pixel by pixel, plot level averages, etc.  There are other questions/concerns as outlined below. Overall, I feel you need to provide more details/clarification on certain steps within your methodology and should try to better connect and clarify the different statistical assessments to your overall goal of evaluating UAV-based RGB VIs for nitrogen assessment. Furthermore, I do feel a more agriculturally focused journal would be a better fit than Sensors as the sensor specific discussions appear to be a minor component of the paper. My comments are below.

Line 100 – It is not clear why VNIR approach would be more expensive than any other UAV-based sensing method given sensor miniaturization, low-cost band pass filters and other methods? This is not true.  A simple RGB camera with a band-bass filter installed to get an NIR-G-B camera is essentially the same cost. Perhaps thermal sensing might be more expensive but a VNIR sensor is likely no more expensive than an RGB camera unless referring to multispectral sensors of higher quality. You need to make this clear and correct this statement.

Line 103- I disagree with this statement. Again, I see no cost benefit to using a RGB camera versus a low-cost consumer grade camera adapted to capture NIR. You need to correct this statement. That is not enough motivation for using RGB cameras alone given the current state of UAS sensor technology and cost.

Line 109 – change to “On the other hand..”

Line 113 – cite references to back this statement

Lines 128-141 – you do not clarify why an alternate color space may or may not be useful for this specific problem of Nitrogen assessment; especially, when describing the CIE-lab space. This needs to be made clear; else this seems superfluous information. Furthermore, your reasons for why you develop a new NDVI like metric from the CIE-lab space needs to be better clarified. The motivation behind it is somewhat vague as written and suggest to try and motivate/explain your reasons better based on limitations, if they exist, for the other RGB-based VIs.

Figure 1 needs a scale bar, north arrow, etc.

Section 2.3 – It is not clear how the RGB images were acquired in the field?  Were they? You stated they were in the introduction. You need to explain here what the two cameras were used for in regard to field data collection and explain the acquisition method used, when were photos taken, how often, how were they georeferenced to align with the UAV imagery, did they occur on the same days as the UAV image acquisition, etc.? Else it is not clear why the vignetting and other corrections are taking place on cameras not used for actua imaging of the plant fields. You state in the next section about field imaging on the ground but don’t really explicitly state it is the same cameras used here. I suggest you reorganize and put image acquisition first before calibration.

Line 237 – I assume you mean CMOS?

Section 2.4 – same statement as above; how were the images georeferenced and coaligned from air and ground?  Another issue/concern is why was only one flying date used to do your analysis? This should be explained.

UAV flight- need to state your overlap used to acquire the imagery and average GSD at the 50 m flying height.

SPAD measurements – these were recorded on different dates relative to the UAV flight – the most recent being two weeks later. Will the two week or greater difference in measurements impact any UAV comparison if using these measurements? If so, this should be in your discussion.

Lines 266-276 – because you are comparing ground acquired RGB images with UAV acquired images, how are these data being co-aligned/georeferenced very precisely to ensure accurate alignment for comparing NDVI measures at the per plant level? No explanation was provided. Else are all the comparisons between ground and UAV restricted to plot averages, not pixel or plant level?

General comment – it appears soil was left in the imagery. Were any attempts to remove soil pixels first performed before computing your VIs?

Section 2.6

Statistical analysis – it is still not clear if you compared plot averages from the UAV data to ground imagery data or pixel to pixel, etc. You need to better clarify and explain your approach. These are very important details and thus far, it is very unclear as to how your derived UAV VI’s are being compared statistically relative to the ground imagery VIs and field measurements. Plant level, plot average,s etc.? And spatial accuracy of the alignment of your measurements is important here too and thus far, no explanation is provided.

Section 3.2 –the paper jumps right into the results here without explaining how these adjustments were tested relative to not correcting for them. Were they? Explain how the VIs were evaluated with and without correction to make clear the purpose and utility of your approach.

Line 406-409 – referring to my comment above, how does the difference in time between SPAD measurements and UAV measurements impact your results? Some discussion on this should be added to the paper.

Lines 424-430 – same comment as above – are your correlations computed from plot averages or pixel to pixel or plant to plant VI values? You need to explain how the values were extracted from each image source and at what scales were these correlations being computed.

Figure 7 appears before reference to Figure 6.

Line 458 – Figure 7 shows not shown

Line 557 – do you mean GGA or GA? GA is not one of the VIs listed as top performing so I’m not sure if you meant GA or GGA, etc.?

Line 600 – “when still developing…”

Line 618 – fix typo – 3parameters

Conclusion –

You need to summarize your study and outline key findings and contributions.

Lines 675-677 – This is a general statement with no evidence that your study achieved this result. How does your study achieve this? It is not clear here in the conclusion unless you explain this statement. You also need some discussion on future work.

Author Response

 We appreciate the reviewer’s comments and have worked to improve the descriptions and details and order of presentation of the Methods, the clarity of the Results and have also rewritten much of our Conclusions with the aim of satisfying the requests for improvement brought forth by the reviewer. Specifically, more details have been added with regards to the scope, meaning and interpretation of the RGB vegetation indices in the Methods and the Results, and the Conclusions have been almost entirely rewritten. Our edits to the manuscript are detailed below in our specific comments in response to each of the issues raised by the reviewer below. As for the submission of this manuscript to the journal Sensors, we have specifically made this submission to the Special Issue: Sensors and Systems for Smart Agriculture. We feel that this is a good location for the publication of the manuscript due to its technical image processing details, new index formulations and data provide on the color calibration techniques used for the different alternate color spaces.

-Line 100 – It is not clear why VNIR approach would be more expensive than any other UAV-based sensing method given sensor miniaturization, low-cost band pass filters and other methods? This is not true.  A simple RGB camera with a band-bass filter installed to get an NIR-G-B camera is essentially the same cost. Perhaps thermal sensing might be more expensive but a VNIR sensor is likely no more expensive than an RGB camera unless referring to multispectral sensors of higher quality. You need to make this clear and correct this statement.

-Line 103- I disagree with this statement. Again, I see no cost benefit to using a RGB camera versus a low-cost consumer grade camera adapted to capture NIR. You need to correct this statement. That is not enough motivation for using RGB cameras alone given the current state of UAS sensor technology and cost.

With regards to the two comments directed at lines 100 and 103, we have eliminated the text at line 100, as we find that it was redundant in light of the reviewer’s comments and have added, at the reviewer’s suggestion, in the part of the line 103 that we are comparing the simple consumer grade RGB cameras, here without modifications, with advanced scientific multispectral VNIR or TIR sensors. Below is the relevant text or changes in text.

It is not our aim in this manuscript or research to conduct a thorough review on the use and potential benefits or limitations of modified RGB cameras, but rather the use of RGB cameras as useful sensors capable of producing meaningful data without physical filter modifications but instead through the use of smart processing techniques using alternate color spaces that allow for the accurate assessment of changes in crop color and fractional vegetation cover.

Line 100-101 “As a low-cost alternative, and at an order of magnitude less expensive than scientific multispectral VNIR or TIR sensors, various Red-Green-Blue Vegetation Indices (RGB-VIs) calculated...”

-Line 109 – change to “On the other hand..”

Below is the relevant text or changes in text.

Line 106 “On the other hand, when these types of camera are used...” 

-Line 113 – cite references to back this statement

We added here the citation for reference 39 (Kefauver et al., 2015), in which several of the already published RGB vegetation indices are compared to NDVI in several different studies; below is the relevant text or changes in text.

Lines 108-110 “Still, RGB-VIs can accurately quantify different properties of color and have often demonstrated performance levels similar to or better than NDVI [39] …”

-Lines 128-141 – you do not clarify why an alternate color space may or may not be useful for this specific problem of Nitrogen assessment; especially, when describing the CIE-lab space. This needs to be made clear; else this seems superfluous information. Furthermore, your reasons for why you develop a new NDVI like metric from the CIE-lab space needs to be better clarified. The motivation behind it is somewhat vague as written and suggest to try and motivate/explain your reasons better based on limitations, if they exist, for the other RGB-based VIs.

We have reviewed the text on these lines and added additional supporting citations and text to the beginning of the paragraph and also further in provided more clarification on the reason for that development the new two indices. The expanded changes in the text are reflected in the Introduction section where indicated at the specified lines and also later on in the Methods section. The supporting justification is two-fold. First, we have added in the Introduction that the previously developed RGB vegetation indices (RGB-VIs) that are part of the Breedpix suite, including GA and GGA, are based on the older Hue, Saturation and Intensity color space, whereas the CIE-Lab and CIE-Luv color spaces are more recent, more advanced and more widely used for color calibration purposes by imaging professionals nowadays. Secondly, these more alternate color spaces are only sparingly used in the previous developments of RGB-VIs as whole image averages of just the single bands of the color space transform, either a*, b*, u* or v*, whereas the full color information is actually combined in two separate bands, with Lightness separate. Here we are attempting to combine the full spectrum of color information with a* and b* as well as u* and v* together and have chosen the normalized difference equation for the specific reasons now explained in further detail in the Methods.

Below is the relevant text or changes in text, with new text highlighted in italics.

In the Introduction, where indicated by the reviewer:

Lines 125-143 “To the end of better quantifying leaf pigment loss, and therefore color changes, due to nitrogen deficiency [46,47], further investigation of the capacities and techniques for accurate color quantification using digital images indicates that there are newer more advanced color models currently in use by photography professionals. In the Commission Internationale de I´Edairage (CIE), CIE-Lab color space model, dimension L* represents lightness; the a* component expresses green to red, with a more positive value representing red, and a more negative value indicating green; and the b* component expresses blue to yellow, where positive values are towards yellow and negative values are closer to blue. Correspondingly, in the CIE-Luv color space model, dimensions u* and v* are perceptually uniform coordinates, where L is again lightness, and u* and v* represent axes similar to a* and b* in separating the color spectrum, respectively. For more specific details on the development of these alternate color space RGB indices and their respective transformations, please see [48]. Both CIE-Lab and CIE-Luv include color calibration corrections through the separation of the color hue from the illumination components of the input RGB signal; for that reason, we have developed two new vegetation indices using these color spaces in a way similar to the conceptual basis for NDVI, using the Normalized Difference between a* and b* (NDLab) and the Normalized Difference between u* and v* (NDLuv). Thus, the CIElab and CIEluv color spaces offer the ability to simultaneously contrast green vegetation quantity with both the reddish/brown soil background (fractional vegetation cover or plant growth) and yellowing caused by chlorosis (loss of foliar chlorophyll), both common symptoms of nitrogen deficiency. Previously, RGB-VI’s have been employed at both the canopy and at the leaf levels for precise crop management or as effective HTPP techniques in breeding programs aimed to improve crop performance under a wide range of conditions [10].” 

In the Methods section:

Lines 341-345 “By inverting a* and u*, more green vegetation becomes a positive contribution to the index while more red/brown soil background reduces the index value. Then, division by b* and v* an increase in yellow chlorotic vegetation will reduce the index. The addition of 1 to the denominator provides for a more balance equation given common values for crops in CIE-Lab and CIE-Luv and normalization then limits the index to values between -1 and 1.”

-Figure 1 needs a scale bar, north arrow, etc.

The Figure 1 was changed to include more geographical information including its context within the country of Zimbabwe and the North Arrow and scale of the phenotyping study. Note that in the overlay image a, we have added a North arrow pointing straight upwards, were as in b, as no geographical registration of the orthomosaic image was conducted, the North arrow points to the left of the image. We felt that this was a better presentation of the data rather than rotating the figure to align with North. We have added the scales to the larger geographical location image and to the inset where the details of the individual microplots are provided as we felt that this was the most relevant scale data. Below is the updated version of Figure 1.

Figure 1. a) Position of the Regional Station of CYMMIT in Harare, Zimbabwe. b) RGB aerial orthomosaic of the 192 plots, with 64 genotypes and 3 replicas per each one, under low managed nitrogen (LOW). c) A plot with specific details of length and width.

-Section 2.3 – It is not clear how the RGB images were acquired in the field?  Were they? You stated they were in the introduction. You need to explain here what the two cameras were used for in regard to field data collection and explain the acquisition method used, when were photos taken, how often, how were they georeferenced to align with the UAV imagery, did they occur on the same days as the UAV image acquisition, etc.? Else it is not clear why the vignetting and other corrections are taking place on cameras not used for actual imaging of the plant fields. You state in the next section about field imaging on the ground but don’t really explicitly state it is the same cameras used here. I suggest you reorganize and put image acquisition first before calibration.

The RGB images were indeed acquired in the field. We had provided information regarding the assessments of the quality of the RGB sensors in the different proposed color spaces first, but we agree with the reviewer’s suggestion to reorganize this part of the manuscript and we have placed first the image acquisition details before calibration. Now the whole of the field data collection follows immediately after Section 2.1 and 2.2 where we have provided the details on the study design and the collection of the relevant agronomic parameters. We have provided below the relevant text or changes in text, starting with 2.3. Apart from the reordering of the RGB image data collection and calibration, new text has been added for clarity regarding some of the more specific comments by the Reviewer, here highlighted in italics.

Lines 218-252

 “2.3 Proximal and aerial data collection

RGB remote sensing evaluations were performed on young maize plants (less than 5 leaves) on January 28th, 2016 during the last week of January. Vegetation indices derived from canopy level image averages per RGB picture taken at the ground level and canopy level whole plot averages of the RGB aerial images. At the ground level, one digital photograph was taken per plot with an Olympus OM-D E-M10 Mark III (Olympus, Tokyo, Japan), holding the camera at about 80 cm above the plant canopy in a zenithal angle and focused near the center of each plot. The images were acquired with a resolution of 16 megapixels with a 4/3'' Live MOS sensor with a focal length of 14 mm, at a speed of 1/125 seconds with the aperture programmed in automatic mode at a resolution of 4608x3072 for a GSD of 0.03 cm/pixel. RGB aerial images were acquired using an UAV (Mikrokopter OktoXL, Moormerland, Germany) flying under manual remote control at 50 m a.g.l. (altitude above ground level). The digital camera used for aerial imaging was a Lumix GX7 (Panasonic, Osaka, Japan), mounted on a two-axis gimbal with vibration reducers for stable image capture while in flight. Images were taken at 16-megapixel resolution of 4592x2448 pixels using a 4/3'' sensor and 20mm focal length lens for an estimated GSD 0.9419 cm/pixel. These images were taken with a 1/160 second shutter speed and auto-programmed mode for maximum aperture at the rate of every 2 s for the duration of the flight and stored locally on microSD cards for subsequent processing.

The measurements of color calibration check and the vignetting calibration were taking the same day of the data collection. We used the ColorChecker Passport Photo (X-Rite, Inc. https://www.xrite.com/es/categories/calibration-profiling/colorchecker-passport-photo/), which has a panel of 24 industry standard color reference chips with published values in RGB as well as the CIE-Lab color space. The photos of this passport were taken with the cameras Olympus OM-D and Lumix GX7 in natural light conditions in a zenithal plane. With the software FIJI (Fiji is Just ImageJ, https://fiji.sc/, https://imagej.nih.gov/ij/), the calibration photos were imported and divided into the separate color channels of Red, Green and Blue and in the CIE-Lab color space as Lightness, a* and b*, and then compared with the 24 published reference values of each standard chip with the photos of the passport taken with the different cameras.

With respect to vignetting calibration, one photo that was taken with the Lumix GX7 at 50 m was divided into the separate RGB and CIE-Lab color space channels. On the R, G, B, Hue, a* and b* single band images, a line was drawn through the center for the X and Y axes in order to extract the cross image transect. Then, a filter was created using the Hue band from the HSI color space in order to select only sunlight soil pixels and applied for R, G, B, a* and b* to eliminate vegetation and shadowed pixel, and the Digital Numbers (DN) were extracted from each remaining point along the line in order to observe changes in albedo across the image axes.”

 -Line 237 – I assume you mean CMOS?

The Camera Olympus OM-D E-M10 Mark III, is described as having a Live MOS Sensor, as described below in the web page of the specific model of digital camera  https://www.olympus.es/site/es/c/cameras/om_d_system_cameras/om_d/e_m10_mark_iii/index.html. We have added in the text the full name of the camera. Further investigation of the Live MOS sensor suggests that it is actually a brand name for a NMOS Image Sensor, which is slightly different than the CCD-based and CMOS based sensors and in some way may provide some of the benefits of each, details that the authors felt were not necessary to the interpretation of this manuscript.

Lines 222-223 “At the ground level, one digital photograph was taken per plot with an Olympus OM-D E-M10 Mark III (Olympus, Tokyo, Japan)." 

-Section 2.4 – same statement as above; how were the images georeferenced and coaligned from air and ground?  Another issue/concern is why was only one flying date used to do your analysis? This should be explained.

For this study the images were not be georeferenced nor coaligned, and no text has been included to suggest as much. This is because we have compared here data analyses of the canopy level image averages per picture taken at the ground level to the canopy level whole plot averages of the UAV images. Indeed, some of the source of the differences between the results from the ground level RGB images and the UAV-acquired aerial level RGB images is that the ground level analyses include a slightly smaller subset of the whole plot, while the UAV RGB analyses are extracted from the orthomosaic image and include 100% coverage of each plot, but that is also considered part of the comparison in the two levels of acquisition. Field level images provide higher resolution and ease of use, but cannot cover the entire plot perfectly, whereas the UAV may provide quicker data collection and complete coverage but require more post-processing. This is discussed later on lines 579-594, copied here for reference.

Lines 579-594: “The RGB-VIs Hue, GGA and NDLab calculated from images taken at ground level demonstrated the best correlations with GY, outperforming other ground RGB indices (Table 3). GA and GGA quantifies the portion of green pixels to the total pixels of the image and is a reliable estimation of vegetation cover [73]. The values of GA from both observation levels (field and aerial) were consistently below 60%. The ground and aerial measurements were taken at the same time on the same day, variation in environmental variables such as light intensity and brightness can be assumed to be negligible. Thus, the main differences must be due to the resolution of the pictures (Figure 2); nevertheless, advances in digital photography allow for sufficiently high resolution for low-altitude aerial imaging to be a viable and economical monitoring tool for agriculture [74]. In this sense, correlations with GY by indices derived from aerial images were generally only slightly weaker than indices measured at ground level. Some of the RGB indices such as NDLab, GGA and GA produced coefficients of determination higher than r=0.75 when comparing the same indices measured from the ground level and from the aerial platform (Table 4). This is despite the methodological differences between index determination at ground level (on an individual plot photo basis) and the aerial platform (mosaicked across a whole trial and further segmented into individual plots) in this case they do not cover the exact same plot area.”

As to the issue of only presenting one flying date used in the analysis, we have discussed that in the Discussion section in section 4.3.2 on lines 595-602 with regards to the general RGB results and again later on at lines 664-685 where we compare these results to other nutrient deficiency studies from the scientific literature, copied below for reference. The single flight data was a limitation of the team, equipment and budget of the project and was aimed for providing the best possible timing for a single acquisition of the study at the earliest possible date that could provide meaningful results for crop phenotyping as this is of interest to the crop breeding community.

Lines 595-602: “This may be best explained considering that the data for our study were collected at an early phenological (vegetative) stage, when the plants were not yet at full canopy cover and they didn't yet show the full range of symptoms of N deficiency, as may be observed in the reproductive stage (Figure 2). N deficiency can reduce plant growth rates, but also other later factors that affect GY, including leaf chlorophyll content, soluble protein content, photosynthetic rate and related enzyme activities of the maize plant during grain filling in the reproductive stage [75-77], which may be a more optimal timing of remote sensing observations when phenotyping low N.”

Lines 664-685: “In a recent study by Gracia-Romero et al. in 2017 [94], the effectiveness of UAVs for canopy level remote sensing for plant phenotyping of maize was similarly demonstrated under different phosphorus nutrient conditions and the results presented therein suggested that the RGB indices were the best option at early growth stages. In the case of low P, however, an equation using GA and u* were the best indicators of GY (R2=0.82). Notwithstanding that their results demonstrated similarly levels of high performance from RGB image analyses, stronger correlations were observed at a similar growth stage of observation, which may be due to the fact that low P symptoms may appear much sooner than low N stress symptoms, presumably due to the higher remobilization capacity of N in plants compared to P, as N absorbed just before flowering is of high importance and has been demonstrated to influence cob growth, the number and size of the grains, and the productivity of the crop [95-97]. Similarly, in a different study P deficiency in soybean plants exhibited higher reflectance in the green and yellow portions of the spectrum but did not show the normal shift in the red edge (related to chlorophyll absorption band at 680 nm) [98]. Even though applied to a different crop in that case, reflectance in the near-infrared (NIR) and blue regions was found to predict early season P stress between growth stages V6 and V8, much earlier than suggested for N deficiencies. With respect to plant N concentrations, the best correlations have been found using reflectance in the red and green regions of the spectrum, while grain yield was best estimated using reflectance in the NIR region, with the wavelengths of importance changing with growth stage (V14-R1) [99,100]. Furthermore, Ma et al., 1996 [101] showed that canopy light reflectance was strongly correlated with field “greenness“(similar to the GA and GGA used in this study) at almost all growth stages, with field greenness in that case being a product of plant leaf area and leaf greenness measured with a chlorophyll meter.”

 -UAV flight- need to state your overlap used to acquire the imagery and average GSD at the 50 m flying height.

We have added the approximate image overlap that we used to build the orthomosaic and have also added the GSD from the 50 m flying height. While the flight height was fixed during the duration of the flight, the image overlap was slightly variable due to manually piloting of the UAV and somewhat windy conditions, hence the specification of “at least” in the overlap. Below is the relevant text or changes in text:

Line 232 “length lens for an estimated GSD 0.9419 cm/pixel with a 1/160 second shutter speed…”

Line 282 “produce an accurate image mosaic with at least 80% overlap and this present…”

 -SPAD measurements – these were recorded on different dates relative to the UAV flight – the most recent being two weeks later. Will the two week or greater difference in measurements impact any UAV comparison if using these measurements? If so, this should be in your discussion.

In the discussion we have explained how the difference in time between SPAD and UAV measurements impacts in the results. While the small difference in timing is not optimal, it is difficult be certain if the two weeks difference between the RGB data acquisition and SPADv made that much of a difference. The results were indeed slightly better when looking at the correlations between SPADv to yield compared to the RGB-VIs, but that is also a more advanced scientific sensor with measurements at the leaf level (thus more time consuming) that include both multispectral absorbance and transmittance. Below is the relevant text for reference.

Lines 635-638 “According to RGB-VIs at both observation levels, the modeled GY estimation was around R2=0.35. That could be a result of the fact that the data capture of RGB indices were taken earlier than the SPADV and SPADR, and at later growth stages the plants may have presented more symptoms related to a lack of nitrogen.

-Lines 266-276 – because you are comparing ground acquired RGB images with UAV acquired images, how are these data being co-aligned/georeferenced very precisely to ensure accurate alignment for comparing NDVI measures at the per plant level? No explanation was provided. Else are all the comparisons between ground and UAV restricted to plot averages, not pixel or plant level?

The data analyses presented here are not pixel or plant level comparisons, but image-level averages either covering a subset of the plot centered on the plot (field level RGB images) or including the whole plot 100% but at lower resolution (UAV aerial RGB images). Below is the relevant new text for reference.

In the Methods section 2.2 on Lines 220-222 “Vegetation indices derived from canopy level image averages per RGB picture taken at the ground level and canopy level whole plot averages of the RGB aerial images.”

In the Methods section 2.5 on Statistical analysis on Lines 350-352. “The results of the canopy level image averages per picture taken at the ground level were compared to the canopy level whole plot averages of the UAV images with Pearson correlation coefficients and ANOVA analyses.”

-General comment – it appears soil was left in the imagery. Were any attempts to remove soil pixels first performed before computing your VIs?

The reviewer is correct in that no specific effort was made to remove the soil background from the imagery. This is because the different RGB vegetation indices either include the fractional amount of vegetation cover as part of their assessments (as in the case of Hue, which measures average plot color) or are specifically measuring fractional vegetation cover of green vegetation (as in the case of GA and GGA, which are the percentages of green + yellowish green pixels or only green pixels in each image). Furthermore, the new indexes NDLab and NDLuv were specifically designed to provide a counterweight of both soil background and yellowish pixels to total green vegetation as now indicated at the end of Section 2.4 in the Methods on lines 341-345: “By inverting a* and u*, more green vegetation becomes a positive contribution to the index while more red/brown soil background reduces the index value. Then, division by b* and v* an increase in yellow chlorotic vegetation will reduce the index. The addition of 1 to the denominator provides for a more balance equation given common values for crops in CIE-Lab and CIE-Luv and normalization then limits the index to values between -1 and 1.”

 Section 2.6

 -Statistical analysis – it is still not clear if you compared plot averages from the UAV data to ground imagery data or pixel to pixel, etc. You need to better clarify and explain your approach. These are very important details and thus far, it is very unclear as to how your derived UAV VI’s are being compared statistically relative to the ground imagery VIs and field measurements. Plant level, plot averages, etc.? And spatial accuracy of the alignment of your measurements is important here too and thus far, no explanation is provided.

We compared canopy level image averages per RGB picture taken at the ground level and canopy level whole plot averages of the RGB aerial images and that was added in different parts of the text. Below is the relevant text or changes in text as added to both section 2.3 on the Proximal and aerial data collection and section 2.5 on Statistical analyses.

Lines 220-222 “Vegetation indices derived from canopy level image averages per RGB picture taken at the ground level and canopy level whole plot averages of the RGB aerial images.”

Lines 350-352 “The results of the canopy level image averages per picture taken at the ground level were compared to the canopy level whole plot averages of the UAV images with Pearson correlation coefficients and ANOVA analyses”

-Section 3.2 –the paper jumps right into the results here without explaining how these adjustments were tested relative to not correcting for them. Were they? Explain how the VIs were evaluated with and without correction to make clear the purpose and utility of your approach.

This section was merged with 3.3 and we have further reorganized and added text to help clarify why we put first the image color and vignetting calibration. Below is the new text that we have added to the manuscript to help further clarify the justification for actions taken.

Lines 418-423: “While the calibration check for the RGB images taken from both cameras demonstrated high correlations (Fig. 5), the result of applying the calibration coefficients to the data resulted in both cases in lower correlations between the vegetation indices at different scales as well as lower correlations between the RGB indices with GY (data not shown). Moreover, the results demonstrated low presence of vignetting effects, with reduced vignetting in the luminescence controlled CIE-Lab and CIE-Luv color spaces of particular interest for this study (Fig. 6).”

-Line 406-409 – referring to my comment above, how does the difference in time between SPAD measurements and UAV measurements impact your results? Some discussion on this should be added to the paper.

In the discussion we have explained how the difference in time between SPAD and UAV measurements may have impacted the results, though, as mentioned earlier, it is difficult to be sure if the differences between SPAD and RGB is due to the date or to the specific type of measurement. Below is the relevant text or changes in text.

Lines 635-638 “According to RGB-VIs at both observation levels, the modeled GY estimation was around R2=0.35. That could be a result of the fact that the data capture of RGB indices were taken earlier than the SPADV and SPADR, and at later growth stages the plants may have presented more symptoms related to a lack of nitrogen.

-Lines 424-430 – same comment as above – are your correlations computed from plot averages or pixel to pixel or plant to plant VI values? You need to explain how the values were extracted from each image source and at what scales were these correlations being computed.

In the Methods sections on image acquisition and processing as well as on the statistical analyses, we have added how the values were extracted from each image source. Below is the relevant text or changes in text.

Lines 220-222 “Vegetation indices derived from canopy level image averages per RGB picture taken at the ground level and canopy level whole plot averages of the RGB aerial images.”

Lines 350-352 “The results of the canopy level image averages per picture taken at the ground level were compared to the canopy level whole plot averages of the UAV images with Pearson correlation coefficients and ANOVA analyses”

-Figure 7 appears before reference to Figure 6.

This was an error of the authors and has been corrected, along with some other reordering of other figures as well as part of some of the improvements in the order of the presentation of the data acquisition and processing and sensor calibration. We changed the Figure 6 to a Figure 2, and now the Figure 2 appears before to the Figure 7.

-Line 458 – Figure 7 shows not shown

 Figure 7 is now presented before the explanation, again this was fixed as part of the larger improvements in the manuscript presentation.

-Line 557 – do you mean GGA or GA? GA is not one of the VIs listed as top performing so I’m not sure if you meant GA or GGA, etc.?

This was an error of the authors and has been corrected. Below is the relevant text or changes in text.

Lines 587-589“GA and GGA quantifies the portion of green pixels to the total pixels of the image and is a reliable estimation of vegetation cover [73].”

 -Line 600 – “when still developing…”

This was an error of the authors and has been corrected. Below is the relevant text or changes in text.

Line 629 "SPADV when still developing roots and leaves may behave as sink organs for the assimilation of N..”

-Line 618 – fix typo – 3parameters

This was an error of the authors and has been corrected. Below is the relevant text or changes in text.

Line 647 “models with a R2 higher than 0.50 included some of the agronomic indicators as estimate parameters.”

Conclusion –

-You need to summarize your study and outline key findings and contributions.

We changed the conclusions with the key findings. Below is the relevant text or changes in text.

Lines 694-703 “Modern phenotyping technologies may help in improving much-needed maize GY in low N conditions, and the current range of variability in performance as indicated by the observed GYLI values suggests that low N and optimal N performance need not be considered mutually exclusive.  For HTPP, RGB sensors can be considered as functional technology with an advanced technology readiness level (TRL) from the ground or a UAV platform, but, similar to the current standard field sensors SPAD and NDVI, the data capture for RGB VIs must be planned accordingly in order to optimize their benefits in support of plant breeding. Several different RGB image-based vegetation indices, including the NDLab and NDLuv indices new to this study, demonstrated similar correlations with GY and contributions to multivariate model GY estimates when compared to standard NDVI and SPAD field phenotyping sensors.”

-Lines 675-677 – This is a general statement with no evidence that your study achieved this result. How does your study achieve this? It is not clear here in the conclusion unless you explain this statement. You also need some discussion on future work.

We added some discussion on future work, below is the relevant text or changes in text.

Lines 703-706 “This study presents possible uses of RGB color image analyses from the ground or from UAVs, with potential benefits compared to currently used field sensors, especially regarding time costs when applied to larger breeding platforms, here demonstrated in application to low N phenotyping in maize.”

We thank the reviewer for their highly constructive criticisms and have worked diligently to provide detailed explanations to their questions and improvements in the manuscript following their suggestions. In some cases, we felt that perhaps the previously presented text had already nearly answered their requests and have provided the supporting text as a reference, in other instances we have made small changes to account for errata but have also made a number of larger changes in others. Some of the criticisms appear to have come from a lack of clarity in a few points on the image processing and we have worked to improve these as much as possible using the reviewer’s suggestions as guidelines for reordering and adding new text as specific places in the manuscript. We thank the reviewer for raising a number of salient points and in making these edits we feel that the manuscript has improved considerably. We are certainly amenable to making further adjustments to the manuscript should there be any other points that require further detail or clarification.
